# Learning from Both Experts and Data

**DOI:** 10.3390/e21121208

**Published:** 2019-12-10

**Authors:** Rémi Besson, Erwan Le Pennec, Stéphanie Allassonnière

**Affiliations:** 1CMAP Ecole Polytechnique, Institut Polytechnique de Paris, 91128 Palaiseau, France; erwan.le-pennec@polytechnique.edu; 2XPop, Inria Saclay, 91120 Palaiseau, France; 3School of Medecine, Université Paris-Descartes, 75006 Paris, France; stephanie.allassonniere@polytechnique.edu

**Keywords:** maximum entropy, mixing expert and data, Kullback–Leibler centroid

## Abstract

In this work, we study the problem of inferring a discrete probability distribution using both expert knowledge and empirical data. This is an important issue for many applications where the scarcity of data prevents a purely empirical approach. In this context, it is common to rely first on an a priori from initial domain knowledge before proceeding to an online data acquisition. We are particularly interested in the intermediate regime, where we do not have enough data to do without the initial a priori of the experts, but enough to correct it if necessary. We present here a novel way to tackle this issue, with a method providing an objective way to choose the weight to be given to experts compared to data. We show, both empirically and theoretically, that our proposed estimator is always more efficient than the best of the two models (expert or data) within a constant.

## 1. Introduction

In this work, we present a novel way to estimate a discrete probability distribution, denoted $p^{🟉}$, using both expert knowledge and data. This is a crucial aspect for many applications. Indeed, when deploying a decision support tool, we often rely entirely on expert/domain knowledge at the beginning; the data only comes with the use of the algorithm in real life. However, we need a good model of the environment to directly train the decision support tool with a planning algorithm. This model of the environment is to be refined and corrected as the data flow increases.

We assume here to have some expert knowledge in the form of an initial a priori on the marginals, the moments, and/or the support of $p^{🟉}$ or any other relevant information. We also assume that we sequentially receive data. We denote $x^{\left(1\right)}$,…, $x^{\left(n\right)}$ an independent and identically distributed (i.i.d) sample following a given unknown discrete probability distribution $p^{🟉}$ in P.

One example of application may come from the objective of building a symptom checker for rare diseases [1]. In this case, $p^{🟉}$ represents the probabilities of the different possible combinations of symptoms, given the event that the disease of the patient is *D*. More precisely, we denote:(1)$$p^{🟉}={\left(p_{1}^{🟉},\ldots ,p_{K}^{🟉}\right)}^{T}=\left(\begin{array}{c} P[\bar{B_{1}},\ldots ,{\bar{B}}_{J-1},\bar{B_{J}}|D]\\ P[\bar{B_{1}},\ldots ,{\bar{B}}_{J-1},B_{J}|D]\\ \vdots \\ P[B_{1},\ldots ,B_{J-1},B_{J}|D] \end{array}\right).$$

We aim to estimate the distribution where *D* is the random variable disease. $B_{1}$,…, $B_{J}$ are the typical symptoms of the disease *D*; all are binary random variables, i.e., the symptom can be present or absent. We aim to estimate the $2^{J}=K$ different combinations (as $P[B_{1},\ldots ,B_{L}|D]$, for example) when we only have an expert a priori on the marginals $P[B_{i}|D]$, for all i∈[1,J].

Of course, a first idea would be to assume that the symptoms are conditionally independent given the disease. However, we expect complex correlations between the typical symptoms of a given disease. Indeed, we can imagine two symptoms that are very plausible individually, but which rarely occur together (or even never, in the case of incompatible symptoms like microcephaly and macrocephaly).

Note also that the assumption of conditional independence would make it possible to present a disease without having any of the symptoms related to this disease in the database (when there is no $B_{i}$ such that $P[B_{i}|D]=1$), which should be impossible.

Generally speaking, if we had enough empirical data, we would no longer need the experts. Conversely, without empirical data, our model must be based entirely on experts. We detail here two different approaches to dealing with the intermediate regime where we do not have enough data to do without the a priori given by the experts, but where we have enough data to correct and specify this initial a priori. These approaches are meaningful as long as we do not know how much data have been used to build the initial a priori, and as long as we really try to combine two heterogeneous forms of information: Experts and empirical data.

In Section 2.1, we first recall the principle of maximum entropy, which is the basic brick we use to build an expert model. We then briefly introduce the proposed approach to mixing experts and data in Section 2.2. We underline the extent to which this approach is superior to the one we previously proposed in [1]. The Barycenter approach that we propose here provides an objective way to choose the weight to be given to experts compared to data. On the contrary, the maximum likelihood with entropic penalization approach of [1] was shown to be sensitive to the choice of the regularization parameter. In Section 3, we outline a review of the literature. Finally, in Section 4, we show both empirically and theoretically that our barycenter estimator is always more efficient than the best of the two models (expert or data) within a constant.

It should be noted that even though we will refer throughout the paper to our particular application in medicine, our framework is relevant for any inference problem involving an initial a priori with a particular form (marginals, moments, support, etc.) combined with data. Biology, ecology, and physics, to name a few, are areas where ideas of maximum entropy have been used for a long time and where the ideas developed in this work could be interesting. See [2] for an overview of the maximum entropy applications for inference in biology.

## 2. Mixing Expert and Empirical Data

### 2.1. Building an Expert Model: The Maximum Entropy Principle

Of course, the aim of benefiting simultaneously from expert data and empirical data has a very old history. This is the very essence of Bayesian statistics [3], which aims to integrate expert data in the form of an a priori, which is updated with empirical data using the Bayes’ theorem to obtain what will be called the posterior.

Note that in our case, we do not have a classical a priori modeling the model parameters with probability distributions. We have an a priori on the marginals, such as a number of constraints on the distribution to be estimated. The absence of an obvious a priori to model the distribution of the parameters naturally leads us to the idea of maximum entropy, theorized by [4]. Indeed, if no model seems more plausible to us than another, then we will choose the least informative. This is a generalization of the principle of indifference often attributed to Laplace:

“We consider two events as equally probable, when we see no reason that makes one more probable than the other, because, even if there is an unequal possibility between them, since we don’t know which is the biggest, this uncertainty makes us look at one as as likely as the other” [5].

This principle therefore takes the form of an axiom that allows us to construct a method to choose an a priori: The least informative possible a priori that is consistent with what we know.

We then define the distribution of maximum entropy as follows:(2)$$p^{\mathrm{maxent}}=\underset{p/p\in \tilde{C}}{arg max}H\left(p\right)$$
where $\tilde{C}=C⋂C^{\mathrm{expert}}$. $C^{\mathrm{expert}}$ is the set of constraints fixed by experts and $C_{K}$ is the probability simplex of the discrete probability distributions of dimension *K*:(3)$$C_{K}=\left\{p=\left(p_{1},\ldots ,p_{K}\right)/\sum_{i=1}^{K}p_{i}=1,p_{i}\geq 0\right\}.$$

Note that $p^{\mathrm{maxent}}$ is well-defined; namely, it exists and is unique, as long as $C^{\mathrm{expert}}$ is a convex set. Indeed, the function p↦H(p) is strictly concave; it is a classic result that a strictly concave function under convex constraints admits a unique maximum.

It is well known that if $C^{\mathrm{expert}}$ only contains the constraints for the marginals, then $p^{\mathrm{maxent}}$ is nothing more than the independent distribution.

However, in our case, we can add some information about the structure of the desired distribution as constraints integrated into $C^{\mathrm{expert}}$. We judge that it is impossible to have a disease without having at least a certain number of its associated symptoms: One, two, or more depending on the disease. Indeed, the diseases we are interested in manifest themselves in combinations of symptoms. The combinations which allow the presence of two simultaneous but exclusive symptoms should also have constraints that are equal to 0. All combinations of constraints are conceivable, as long as $\tilde{C}$ remains a convex closed space, in order to ensure the existence and uniqueness of $p^{\mathrm{maxent}}$.

We therefore construct our a priori by taking the maximum entropy distribution, checking the constraints imposed by the experts. Thus, among the infinite distributions that verify the constraints imposed by the experts, we choose the least informative distribution $p^{\mathrm{maxent}}$; in other words, the one closest to the conditional independence distribution.

We need to add information to move from the information provided by the experts to the final distribution, and we want to add as little as possible to what we do not know. This approach is referred to as maxent (maximum entropy) and has been widely studied in the literature [4,6,7].

### 2.2. Barycenters between Experts and Data

Our target probability distribution is denoted $p^{🟉}=\left(p_{1}^{🟉},\ldots ,p_{K}^{🟉}\right)$ and is defined on the probability simplex $C_{K}$ of Equation (Equation 3).

The i.i.d. sample of $p^{🟉}$ is denoted $x^{\left(1\right)}$,…, $x^{\left(n\right)}$, where $x^{\left(i\right)}\in R^{K}$. The empirical distribution $p_{n}^{\mathrm{emp}}={\left(p_{n,i}^{\mathrm{emp}}\right)}_{i=1}^{K}$ is given by:(4)$$p_{n,i}^{\mathrm{emp}}=\frac{1}{n}\sum_{j=1}^{n}1_{\left\{x^{\left(j\right)}=i\right\}}.$$

Following the ideas of Section 2.1, we define the expert distribution as the distribution which maximizes entropy while satisfying the constraints fixed by experts:(5)$$p^{\mathrm{expert}}=\underset{p/p\in \tilde{C}}{arg max}H\left(p\right)$$
where $\tilde{C}$ is the intersection of the simplex probabilities with the set of constraints fixed by experts. In our medical context, the set of constraints is composed of a list of censured combinations and a list of marginals coming from the literature. Censured combinations are combinations of symptoms that are set to zero because they involve the simultaneous presence of two incompatible symptoms and/or combinations that do not involve enough of a presence of typical symptoms.

Note that it is possible to give more or less credit to the marginals given by experts by formulating the constraint as an interval (wider or narrower) rather than as a strict equality. The distribution of experts is then defined as the least informative distribution consistent with what we know.

Let L be any dissimilarity measured between two probability distributions. Our barycenter estimator mixing expert and empirical data is then defined as:(6)$${\hat{p}}_{\epsilon_{n}}^{L}=\underset{p\in C/L\left(p_{n}^{\mathrm{emp}},p\right)\leq \epsilon_{n}}{arg min}L\left(p^{\mathrm{expert}},p\right)$$
where
(7)$$\epsilon_{n}:=\epsilon_{n}^{\delta }=\underset{l}{arg min}P\left[L\left(p_{n}^{\mathrm{emp}},p^{🟉}\right)\leq l\right]\geq 1-\delta$$
and P is the probability measure defined on the product space $\{(x^{\left(1\right)}$,…, $x^{\left(n\right)})∼{⊗}_{i=1}^{n}p^{🟉};n\geq 1\}$.

${\hat{p}}_{n}^{L}$ is then defined as the closest distribution from the experts, in the sense of the dissimilarity measure L, which is consistent with the observed data.

For such a construction to be possible, we will therefore have to choose a measure of dissimilarity L so that we have a concentration of the empirical distribution around the true distribution for L.

Such a formulation has several advantages over the maximum likelihood with entropic penalization approach previously proposed in [1]. First, we do not have to choose a regularization parameter, which seems to have a strong impact on the results of the estimator (see [1]). This parameter is replaced by the parameter δ, for which it is reasonable not to take more than 0.1 and which appears to have low impact on the result of ${\hat{p}}_{n}^{L}$ (see Section 5). Secondly, the solution of (Equation 6) can be (depending on the choice of the dissimilarity measure L) easier to compute than that of the optimization problem associated with the penalization approach, for which a closed form of the solution could not be derived [1]. Mathematically, ${\hat{p}}_{n}^{L}$ is the projection of the experts on the confidence interval centered on the empirical distribution and radius $\epsilon_{n}$. Figure 1 and Figure 2 give a visual interpretation of such a construction. These representations should not be taken literally. The objects we work on live in the simplex of probabilities, and their geometry is very different from the Euclidean interpretation of Figure 1 and Figure 2. These figures are mainly here to illustrate the two different cases we can have. In Figure 1, we do not have much data and the confidence interval is wide. In this case, the projection of the experts on the confidence interval centered on the empirical distribution is the expert distribution itself. We do not have enough elements to modify our initial a priori. This case can also occur when the initial a priori of the experts is very close to the true distribution. On the contrary, in Figure 2, we modify the initial a priori because the experts do not belong to the confidence interval anymore.

### 2.3. A Baseline: Mixture Model with Bayesian Weights

We present here an alternative approach that will be our baseline in the numerical experiments of Section 5. We still aim to mix the empirical distribution, $p_{n}^{\mathrm{emp}}$, built with an i.i.d. sample of $p^{🟉}$: $x^{\left(1\right)}$,…, $x^{\left(n\right)}$, with the distribution of the experts $p^{\mathrm{expert}}$.

The idea is to make a linear combination of these two models:$${\hat{p}}_{n}^{\mathrm{Bayes}}\propto \gamma_{1}p_{n}^{\mathrm{emp}}+\gamma_{2}p^{\mathrm{expert}}$$
where the mixture parameters are proportional to the log-likelihood of the data according to the model considered, namely:$$\gamma_{1}=\ell \left(x^{\left(1\right)},\ldots ,x^{\left(n\right)}|p_{n}^{\mathrm{emp}}\right)$$
and
$$\gamma_{2}=\ell \left(x^{\left(1\right)},\ldots ,x^{\left(n\right)}|p^{\mathrm{expert}}\right)$$
where *ℓ* stands for the log-likelihood.

This is a parametric Bayesian approach, since we apply the Bayes theorem, stating that the posterior is proportional to the product of the prior with the likelihood function.

## 3. Related Works

### 3.1. Expert System with Probabilistic Reasoning

The creation of a decision support tool for medical diagnosis has been an objective since the beginning of the computer age. Most of the early work proposed a rule-based expert system, but in the 1980s, a significant part of the community studied the possibility of building an expert system using probabilistic reasoning [8]. Bayesian probabilities and methods were therefore considered as good ways to model the uncertainty inherent in medical diagnosis relatively early.

The assumption of conditional independence of symptoms given the disease has been intensively discussed, as it is of crucial importance for computational complexity. Some researchers considered this hypothesis harmless [9], while others already proposed a maximum entropy approach to face this issue [10,11,12].

However, it seems that none of the work of that time considered the expert vs. empirical data trade-off that we face. In the review article [13] presenting the state-of-the-art of the research of that time (1990) about this issue, it is clearly mentioned that these methods only deal with data of probabilistic forms. More precisely, they assume that they have an a priori on the marginal but also on some of the combinations of symptoms (in our case, we would assume that we have an a priori on $P[B_{1},B_{2}|D]$, for example), and propose a maximum entropy approach where these expert data are treated as constraints in the optimization process. Once again, this is not the case for us, since we have only an a priori on the marginal (and a certain number of constraints), as well as experimental data. This field of research was very active in the 1980s and then gradually disappeared, probably due to the computational intractability of the algorithms proposed for the computer resources of the time.

### 3.2. Bayesian Networks

Bayesian networks [14] were then quickly considered as a promising alternative for modeling probabilistic dependency relationships between symptoms and diseases [8]. These are now used in most expert systems, particularly in medicine [15].

A Bayesian network is generally defined as an acyclically oriented graph. The nodes in this graph correspond to the random variables: Symptoms or diseases in our case. The edges link two correlated random variables by integrating the information of the conditional law of the son node with respect to the father node. The main advantage of such a model is that it can factorize the joint distribution using the so-called global Markov property. The joint law can indeed be expressed as the product of the conditional distributions of each node given its direct parents in the graph [16].

First of all, the construction of a Bayesian network implies the inference of its structure, i.e., to determine the nodes that must be linked by an edge to those that can be considered conditionally independent of the rest of the graph (structure learning). Then, learning the network implies learning the parameters, i.e., the probabilities linking the nodes (parameter learning).

It is therefore natural to also find works that aimed at mixing expert and empirical data in this area of the literature. In [17], the experts’ indications take a particular form since they indicate by hand correlations, positive or negative, between variables. The approach of [18] is also quite distant, because it is prefers being based on data. [18] only uses expert indications for additional variables for which there are no data, typically rare events never observed in the database. A work closer to ours is [19], where the authors assume that they have a first Bayesian network built entirely by the experts, to which they associate a degree of trust. The authors then use the available data to correct this expert network. We distinguish ourselves from this work in our effort to find an objective procedure for the weight to be given to experts in relation to the data (and for this weight not to be set by the experts themselves).

Note also that the main interest of Bayesian networks is to take advantage of conditional independence relationships known in advance, as they are pre-filled by experts or inferred from a sufficient amount of data. However, in our case, we do not have such an a priori knowledge about the dependency relationships between symptoms and or enough data to infer them.

### 3.3. From the Marginals to the Joint Distribution

Estimating the joint distribution from the marginal is an old problem, which is obviously not necessarily related to expert systems. This problem is sometimes referred to in the literature as the “cell probabilities estimation problem in contingency tables with fixed marginals”. The book [20] gives a good overview of this field. We can trace back to the work of [21], which assumes knowing the marginal and having access to a sample of empirical data, and aims to estimate the joint distribution. In this article, they proposed the “iterative proportional fitting procedure” (IPFP) algorithm, which is still very popular for solving this problem.

An important assumption of [21] is that each cell of the contingency table receives data. In [22], the authors prove that the asymptotic estimator obtained by an IPFP algorithm is the distribution that minimizes the Kullback–Leibler divergence from the empirical distribution under the constraint to respect the marginal experts.

However, an IPFP algorithm is not suitable for our problem for two main reasons: First, we do not have absolute confidence in the marginals given by experts (we want to allow ourselves to modify them as we collect more data) and second, because we are interested in rare diseases, we do not expect to have a sufficient amount of data. In fact, many of the cells in the contingency table we are trying to estimate will not receive data, but it would be disastrous in our application to assign a zero probability to the corresponding symptom combination.

In a sense, an IPFP algorithm does exactly the opposite of what we are aiming for: It modifies empirical data (as little as possible) to adapt them to experts, while we aim to modify experts (as little as possible) to make them consistent, in a less restrictive sense, with empirical data.

We should also mention the work related to our problem in applications of statistics in the social sciences, where researchers aim to construct a synthetic population from the marginal, coming from several inconsistent sources [23]. Their proposed approach also uses ideas of maximum entropy, but it is still different from our trade-off of expert vs. empirical data, since they built their model without samples.

### 3.4. The Kullback Centroid

Our optimization problem in Equation (Equation 6) in the particular case where the dissimilarity measure L is the Kullback–Leibler divergence is called moment-projection (M-projection) in the literature. The properties of these projections have been intensely studied [24].

Note that the Lagrangian associated with such an optimization problem is then nothing more than a Kullback–Leibler centroid. These objects or variations/generalizations of them (with Jeffrey’s, Bregman’s divergences, etc.) have been the subject of research since the paper of [25]. For example, articles [26,27] study cases where an exact formula can be obtained, and propose algorithms when this is not the case.

However, we have not found any use of these centroids to find a good trade-off of expert vs. empirical data as we propose in this paper. Bregman’s divergence centroids have been used to mix several potentially contradictory experts; the interested reader may refer to the work of [28,29]. We could certainly consider that the empirical distribution $p_{n}^{\mathrm{emp}}$ is a second expert, and that our problem is the same as mixing two experts: Literature and data. However, the question of the weight to be given to each expert, which is the question that interests us here, will not be resolved. In [28], the aim is rather to synthesize contradictory opinions of different experts by fixing the weight to be given to each expert in advance. We propose, for our part, an objective procedure to determine the weight to be given to experts compared to empirical data.

## 4. Theoretical Properties of the Barycenter Estimator

### 4.1. Barycenter in $L^{p}$ Space

In this section we work in the $L^{p}$ space. Let us recall that the classic norm on the space $L^{p}$ is given by: ${∥x∥}_{p}={\left(\sum_{i}{\left|x_{i}\right|}^{p}\right)}^{\frac{1}{p}}$.

Following the ideas presented in Section 2.2, we define our estimator, ∀i≥1 as follows:(8)$${\hat{p}}_{n}^{i}=\underset{p\in C/{∥p-p_{n}^{\mathrm{emp}}∥}_{i}\leq \epsilon_{n}}{arg min}{∥p-p^{\mathrm{expert}}∥}_{i}$$
where
(9)$$\epsilon_{n}:=\epsilon_{n}^{\delta }=\underset{l}{arg min}P\left[{∥p_{n}^{\mathrm{emp}}-p^{🟉}∥}_{i}\leq l\right]\geq 1-\delta .$$

To control $\epsilon_{n}$, we use the concentration inequality obtained in the recent work of [30]. In the literature, most of the concentration inequalities for the empirical distribution use the $L^{1}$ norm. This is why, even though we will present the following results by trying to generalize to spaces $L^{p}$ for all p, in practice, only ${\hat{p}}_{n}^{1}$ interests us. The proofs for the different theoretical results of this section are relegated to the Appendix A.

**Proposition** **1**(Existence and uniqueness). *The estimator ${\hat{p}}_{n}^{i}$ defined by *(Equation 8)* exists for all i≥1.*
*${\hat{p}}_{n}^{i}$ is unique if and only if i≠1.*

*In the following, ${\hat{p}}_{n}^{1}$ therefore refers to a set of probability measures.*


**Proof.** See Section A.1. □

The next proposition shows that one of the solutions of (Equation 8) can always be written as a barycenter between $p_{n}^{\mathrm{emp}}$ and $p^{\mathrm{expert}}$. This property therefore provides us with an explicit expression of a solution of (Equation 8), which was not otherwise trivial to obtain by a direct calculation looking for the saddle points of the Lagrangian (for example, in the case i=1).

**Proposition** **2.**
*Let ${\hat{p}}_{n}^{i}$ be defined by *(Equation 8)*; then for all i, it exists $\tilde{p}\in {\hat{p}}_{n}^{i}$ such that $\exists \alpha_{n}\in \left[0,1\right]$:*
(10)$$\tilde{p}=\alpha_{n}p^{\mathrm{expert}}+\left(1-\alpha_{n}\right)p_{n}^{\mathrm{emp}}$$
*where $\alpha_{n}=\frac{\epsilon_{n}}{{∥p_{n}^{\mathrm{emp}}-p^{\mathrm{expert}}∥}_{i}}$ if $\epsilon_{n}\leq {∥p_{n}^{\mathrm{emp}}-p^{\mathrm{expert}}∥}_{i}$ and $\alpha_{n}=1$ otherwise.*


**Proof.** See Section A.2. □

Therefore, one of the elements of ${\hat{p}}_{n}^{1}$ can be written under the form of a barycenter. For the sake of simplicity, in the following, we will designate ${\hat{p}}_{n}^{1}$ as the solution of (Equation 8) for i=1, which can be written under the form of (Equation 10) rather than using the whole set of solutions.

It is now a question of deriving a result proving that mixing experts and data, as we do with ${\hat{p}}_{n}^{1}$, represents an interest rather than a binary choice of one of the two models. For this reason, we show in the following proposition that, with a high probability, our estimator ${\hat{p}}^{1,1}$ is always better than the best of the models within a constant.

**Theorem** **1.**
*Let ${\hat{p}}_{n}^{1}$ be defined by *(Equation 8)*. Then, we have with probability of at least 1−δ:*
(11)$${∥p^{🟉}-{\hat{p}}_{n}^{1}∥}_{1}\leq 2\min\left\{\epsilon_{n},{∥p^{🟉}-p^{\mathrm{expert}}∥}_{1}\right\}.$$


**Proof.** See Section A.3. □

### 4.2. Barycenter Using the Kullback–Leibler Divergence

In this section, we study the theoretical properties of the solution of Equation (Equation 6) in the particular case where the dissimilarity measure L is the Kullback–Leibler divergence. The proofs for the different theoretical results of this section are relegated to the Appendix B.

The Kullback–Leibler divergence between two discrete probability measures *p* and *q* is defined as:$$\mathrm{KL}\left(p||q\right)=\sum_{i}p_{i}\log\left(\frac{p_{i}}{q_{i}}\right).$$

Let us recall that the Kullback-Leibler divergence is not a distance, since it is not symmetric and does not satisfy the triangular inequality; however, it is positively defined [6].

We define our estimator as:(12)$${\hat{p}}_{n}^{L}=\underset{p\in C/\mathrm{KL}(p_{n}^{\mathrm{emp}}\left||p\right)\leq \epsilon_{n}}{arg min}\mathrm{KL}(p^{\mathrm{expert}}\left||p\right)$$
where
(13)$$\epsilon_{n}:=\epsilon_{n}^{\delta }=\underset{l}{arg min}P[\mathrm{KL}(p_{n}^{\mathrm{emp}}\left|\right|p^{🟉})\leq l]\geq 1-\delta .$$

To calibrate $\epsilon_{n}$, we can use the concentration inequality obtained in [30]. More precisely, we have:(14)$$\epsilon_{n}=\frac{1}{n}\left(-\log\left(\delta \right)+\log\underset{=:G_{n}}{\underset{︸}{\left(3+3\sum_{i=1}^{K-2}{\left(\sqrt{\frac{e^{3}n}{2\pi i}}\right)}^{i}\right)}}\right).$$

In the following proposition, we show the existence and uniqueness of our estimator ${\hat{p}}_{n}^{L}$ and the fact that our estimator is a barycenter. However, unlike in the case of ${\hat{p}}_{n}^{1}$ of Equation (Equation 8), it does not seem possible to obtain a closed form for ${\hat{p}}_{n}^{L}$ this time.

**Proposition** **3.**
*Let ${\hat{p}}_{n}^{L}$ be defined by *(Equation 12)*; then, ${\hat{p}}_{n}^{L}$ exists and is unique. Moreover, ${\hat{p}}_{n}^{L}$ can be written under the following form:*
(15)$${\hat{p}}_{n}^{L}=\frac{1}{1+\tilde{\lambda }}p^{\mathrm{expert}}+\frac{\tilde{\lambda }}{1+\tilde{\lambda }}p_{n}^{\mathrm{emp}}$$
*where $\tilde{\lambda }$ is a non-negative real such that:*
(16)$$\tilde{\lambda }\geq \frac{\mathrm{KL}(p_{n}^{\mathrm{emp}}||p^{\mathrm{expert}})}{\epsilon_{n}}-1.$$


**Proof.** See Section B.1. □

The following proposition is intended to be the analog of the proposition 1 when L is the Kullback–Leibler divergence. We prove that the centroid ${\hat{p}}_{n}^{L}$ is better than the experts (with high probability). On the other hand, we obtain that when $\mathrm{KL}(p_{n}^{\mathrm{emp}}\left||\right|p^{🟉})>\mathrm{KL}(p^{\mathrm{expert}}\left|\right|p^{🟉})$, the ${\hat{p}}_{n}^{L}$ barycenter is better than the empirical distribution. To obtain guarantees when $\mathrm{KL}(p_{n}^{\mathrm{emp}}\left|\right|p^{🟉})\leq \mathrm{KL}(p^{\mathrm{expert}}\left|\right|p^{🟉})$ seems less obvious and requires control over the quantity $\mathrm{KL}(p_{n}^{\mathrm{emp}}||p^{\mathrm{expert}})$.

**Theorem** **2.**
*Let ${\hat{p}}_{n}^{L}$ be defined by *(Equation 12)*; then, we have with probability at least 1−δ:*
(17)$$\mathrm{KL}({\hat{p}}_{n}^{L}\left|\right|p^{🟉})\leq \min\left\{\mathrm{KL}(p^{\mathrm{expert}}\left|\right|p^{🟉}),\epsilon_{n}\left(L_{n}+1\right)\right\}$$
*where*
$$L_{n}=\frac{\mathrm{KL}(p^{\mathrm{expert}}\left|\right|p^{🟉})-\mathrm{KL}(p_{n}^{\mathrm{emp}}\left|\right|p^{🟉})}{\mathrm{KL}(p_{n}^{\mathrm{emp}}||p^{\mathrm{expert}})}.$$


**Proof.** See Section B.2. □

**Remark** **1.**
*Note that $\mathrm{KL}({\hat{p}}_{n}^{L}||p^{🟉})$ is infinite if $p^{\mathrm{expert}}$ does not have the same support as that of $p^{🟉}$. Nevertheless, obtaining a result for $\mathrm{KL}(p^{🟉}||{\hat{p}}_{n}^{L})$ would require us to have a concentration on $\mathrm{KL}(p^{🟉}||{\hat{p}}_{n}^{\mathrm{emp}})$, which we do not have. Note that $\mathrm{KL}(p^{🟉}||{\hat{p}}_{n}^{\mathrm{emp}})$ is infinite until we have sampled all the elements of the support of $p^{🟉}$ at least one time.*


## 5. Numerical Results

For each experiment in this section, we generate a random distribution $p^{🟉}$ that we try to estimate. To do this, we simulate some realizations of a uniform distribution and renormalize in order to sum up to 1.

We also generate four different distributions that will serve as a priori for the inference: $p^{\mathrm{expert},i},\forall i\in \left\{1,2,3,4\right\}$. The first three priors are obtained by a maximum entropy procedure under constraint to respect marginals of $p^{🟉}$ having undergone a modification. We added to the marginals of $p^{🟉}$ a Gaussian noise of zero expectation and variance equal to $\sigma_{1}^{2}=0.1$, $\sigma_{2}^{2}=0.2$ and $\sigma_{3}^{2}=0.4$, respectively. The last prior $p^{\mathrm{expert},4}$ is chosen to be equal to the distribution $p^{🟉}$ (the experts provided us with the right distribution).

We then sequentially sample data from $p^{🟉}$, i.e., we generate patients, and update for each new datum and each different a priori, the left centroid ${\hat{p}}_{n}^{L}$ (using an Uzawa algorithm [31]), the barycenter ${\hat{p}}_{n}^{1,1}$, and the empirical distribution $p_{n}^{\mathrm{emp}}$, as well as the divergences $\mathrm{KL}({\hat{p}}_{n}^{L}||p^{🟉})$ and $\mathrm{KL}(p_{n}^{\mathrm{emp}}||p^{🟉})$ and the norms ${∥{\hat{p}}_{n}^{1,1}-p^{🟉}∥}_{1}$ and ${∥p_{n}^{\mathrm{emp}}-p^{🟉}∥}_{1}$.

The experiments of Figure 3, Figure 4, Figure 5 and Figure 6 were conducted on the case of a disease with J=7 typical symptoms and where there are therefore $K=2^{7}=128$ possible combinations. The experiments of Figure 7, Figure 8 and Figure 9 were conducted on the case of a disease with 9 typical symptoms and where there are therefore $K=2^{9}=512$ possible combinations.

### 5.1. General Analysis of the Barycenter Performance and Choice of $\epsilon_{n}$

The only parameter that we can control is the δ used to construct the confidence interval of the concentration of the empirical distribution around the true distribution. Let us recall that for the case of the Kullback centroid of Equation (Equation 12), we set:(18)$$\epsilon_{n}=\frac{1}{n}\left(-\log\left(\delta \right)+\log\left(G_{n}\right)\right)$$
where $G_{n}$ is defined in Equation (Equation 14).

However, our first numerical experiments show that the choice of $\epsilon_{n}$ defined by Equation (Equation 18) is a little too conservative (see Figure 3). We need to converge $\epsilon_{n}$ faster towards 0 without abandoning our a priori when it is good.

Our experiments suggest taking an $\epsilon_{n}$ consistent with the concentration proposed in a conjecture of [30] for Kullback–Leibler divergence:(19)$$\epsilon_{n}=\frac{-\log\left(\delta \right)+\frac{n}{2}\log\left(1+\frac{K-1}{n}\right)}{n}.$$

Note that we added a constant $\frac{1}{2}$ to the conjecture of [30]. As for the choice of δ, this appears important mainly when *n* is small; taking it when sufficiently low avoids an overfitting situation when the number of data is still low, without being harmful when *n* is high. We took it equal to $10^{-6}$ in the experiments of Figure 3, Figure 4 and Figure 5 and Figure 7, and tried different values in Figure 6.

The Figure 5 and Figure 7 show that such a choice for $\epsilon_{n}$ makes a good trade-off between expert and empirical data, because we are able to take advantage of these two sources of information when the number of data is small (typically when n<K), but also to quickly abandon our a priori when it is bad (see the black curves) or to keep it when it is good (the green curves). Eventually, the Figure 5 and Figure 7 were performed on problems of 128 and 512, respectively, and this choice of $\epsilon_{n}$ therefore appears relatively robust with respect to changes in size.

Concerning ${\hat{p}}_{n}^{1.1}$, we took, still following the conjectures of [30]:(20)$$\epsilon_{n}=\sqrt{\frac{-\log\left(\delta \right)+\frac{n}{2}\log\left(1+\frac{K-1}{n}\right)}{n}}.$$

The Figure 4 shows the error made by our barycenter in norm $L^{1}$: ${\hat{p}}_{n}^{1}$ using such an $\epsilon_{n}$. We are again able to get rid of a bad a priori relatively quickly to follow the empirical (green curve) while keeping it if it is good (blue curve).

Moreover, we show with these experiments that there is an intermediate regime when we do not have much data, where our estimator is *strictly* better than the two individual models (experts and data alone). This is particularly visible when we used the $\epsilon_{n}$ of the conjecture of [30] (see Figure 7 and Figure 5). It is then empirically evident that mixing these two heterogeneous sources of information, experts and empirical data, can be useful for statistical inference.

One might nevertheless wonder by looking at the experiments of Figure 3, Figure 4 and Figure 5 and Figure 7 why we propose a mixture of expert and data rather than just a binary choice of the best model. Indeed, both our theoretical and experimental results show that we can lose a constant when making a barycenter between expert and data instead of just a binary choice of the best of the two models. This is particularly true when the number of data tends to grow and when the initial expert a priori was misleading. Nevertheless, this constant is a price that we are willing to pay in order to avoid the undesirable consequences of a binary choice of model.

First, when making a binary choice of model, it is not that easy to determine when we should jump from the expert model to the empirical model. Note also that it would produce an undesirable discontinuity in the function of the data flow. Most importantly, it is crucial in our application that our estimator has the same support as the real distribution. It would be disastrous indeed to consider that a disease is impossible because we never observed a particular combination of symptoms. This remark is somewhat linked to the well-known coupon collector’s problem: How many samples do we need on average to observe all the modalities of the support of a given distribution at least one time? In the equal case (the target distribution is uniform), the average number of samples needed is of the order of Klog(K), but it might be much more in the unequal case [32]. Nevertheless, let us emphasize here once again that we are particularly interested in the moment where we have little data. Then, the empirical distribution alone will never be a really good alternative. We could, of course, consider a Laplace smoothing in order to avoid this difficulty, but this would be nothing more than a less sophisticated maximum entropy approach.

### 5.2. Comparison with the Baseline and Choice of δ

In Figure 6, Figure 8 and Figure 9 we compare our approach with the Bayesian mixture of Section 2.3. We removed the empirical distribution curve for visual reasons, because it is always above the curves presented and thus distorts the representation by stretching the y-axis.

We tried two different values for our only parameter δ: $\delta =10^{-1}$ and $\delta =10^{-6}$. Note that the advantage of our method is that the parameter that we have to choose, δ, has an intelligible meaning: It refers to the probability that $p^{🟉}$ is outside the confidence interval. That is why we do not consider higher values of δ.

First of all, one can note the influence of the δ parameter on Figure 6, Figure 8 and Figure 9. The light yellow curve is a clear example of where the δ has been chosen too high, $10^{-1}$, giving too much credit to data in comparison with the expert. There is of course a trade-off; to choose a smaller δ, $10^{-6}$ has a cost, as we can see with the black and the dark blue curves, which are a bit too conservative in comparison with the dark green and the dark yellow ones.

Nevertheless, despite the variability of the observed performance of our method as a function of δ, it leads in any case to a better mixture than the baseline in all our experiments. Our barycenter then outperforms the baseline in this task of finding the right weight to give to data in relation to the expert. This is particularly true when $\delta =10^{-6}$ and to a lesser extent when $\delta =10^{-1}$.

Indeed, we aim at finding a mixture that would keep the expert a priori when it is good and quickly move away when it is bad. This is not what the baseline exhibits in our experiments, contrary to our estimator. The light green curve shows clearly that the weight placed on the data in relation to the expert is too high; the beginning of the purple curve also exhibits this behavior.

Once again, the Figure 8 shows that such observations are robust with respect to changes of dimension, and the Figure 8 has $2^{9}=512$ symptom combinations; meanwhile, the Figure 6 has $2^{7}=128$.

## 6. Conclusion and Perspectives

In this work, we have presented a way to combine expert knowledge—in the form of marginal probabilities and rules—together with empirical data in order to estimate a given discrete probability distribution. This problem emerged from our application, in which we aimed to learn the probability distribution of the different combinations of symptoms of a given disease. For this objective, we have an initial a priori consisting of the marginal distributions coming from the medical literature; clinical data collected is used as the decision support tool.

The particular form of the prior does not allow us to simply adopt a maximum a posteriori (MAP) approach. The absence of an obvious a priori to model the parameter’s distribution naturally leads us to the idea of maximum entropy: If no model seems more plausible to us than another, then we will choose the least informative.

This idea of maximum entropy brings us back to the works of the 1980s and 1990s, where researchers also aimed to build symptom checkers using marginals. In our work, we go further by gradually integrating empirical data as the algorithm is used.

We are interested in the intermediate regime in which we do not have enough empirical data to do without experts, but have enough to correct them if necessary. Our proposal is to construct our estimator as the distribution closest to the experts’ initial a priori, in the sense of a given dissimilarity measure, that is consistent with the empirical data collected.

We prove, both theoretically and empirically, that our barycenter estimator mixing the two sources of information is always more efficient than the best of the two models (clinical data or experts alone) within a constant.

We have empirically illustrated the effectiveness of the proposed approach by giving an a priori of different quality and incrementally adding empirical data. We have shown that our estimator allows a bad a priori to be abandoned relatively quickly when the inconsistency of the data collected with the initial a priori is observed. At the same time, this same mixture makes it possible to keep the initial a priori if it is good. Moreover, we show with this experiment that, in the intermediate regime, our estimator can be *strictly* better than the best of the two models (experts and data alone). It empirically confirms the idea that mixing these two heterogeneous sources of information can be profitable in statistical inference.

Future work will concentrate on several refinements, such as the addition of a kernel structure for the construction of the empirical distribution. Indeed, it is possible that there are omissions of some symptoms in the data collected. Then, a kernel approach that would consider closer states that only differ by some presences would capture such a difficulty and make a better use of empirical data. Other dissimilarity measures could also be investigated. Finally, having a true non-parametric Bayesian approach would be very interesting. However, closing the gap between classical Dirichlet priors on the marginal to a single prior on the joint distribution seems to be a real challenge.

## Figures and Tables

**Figure 1 entropy-21-01208-f001:**
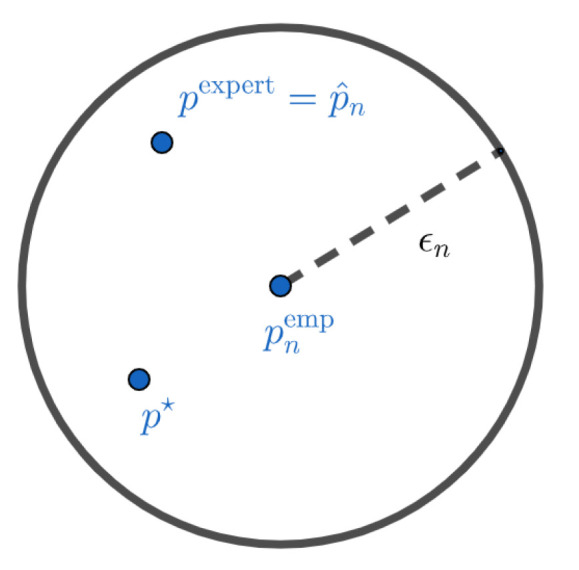
Barycenter between expert and data when the expert belongs to the confidence interval centered in the empirical distribution. In this case, there is no sufficient empirical evidence that the expert is wrong.

**Figure 2 entropy-21-01208-f002:**
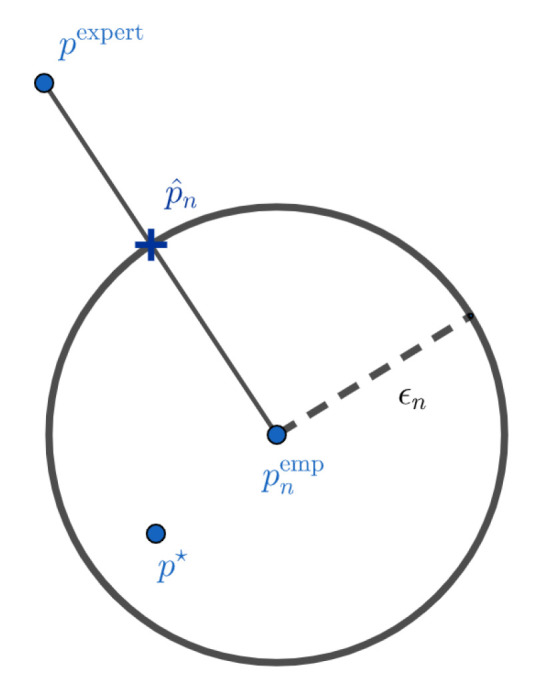
Barycenter between expert and data when the expert does not belong to the confidence interval centered in the empirical distribution. There is a high probability that the expert is outside the set where the target is located and therefore needs to be corrected.

**Figure 3 entropy-21-01208-f003:**
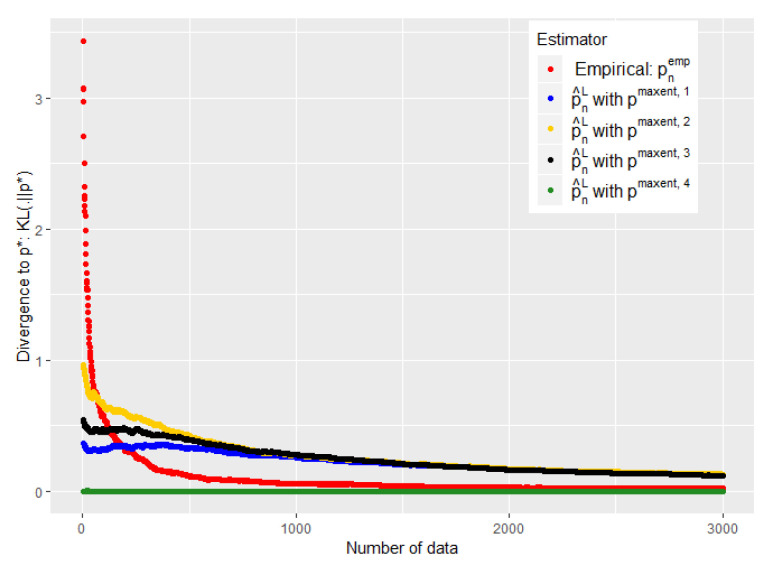
Evolution of the performance of ${\hat{p}}_{n}^{L}$ as a function of the available number of empirical data. $\epsilon_{n}$ is defined by Equation (Equation 18).

**Figure 4 entropy-21-01208-f004:**
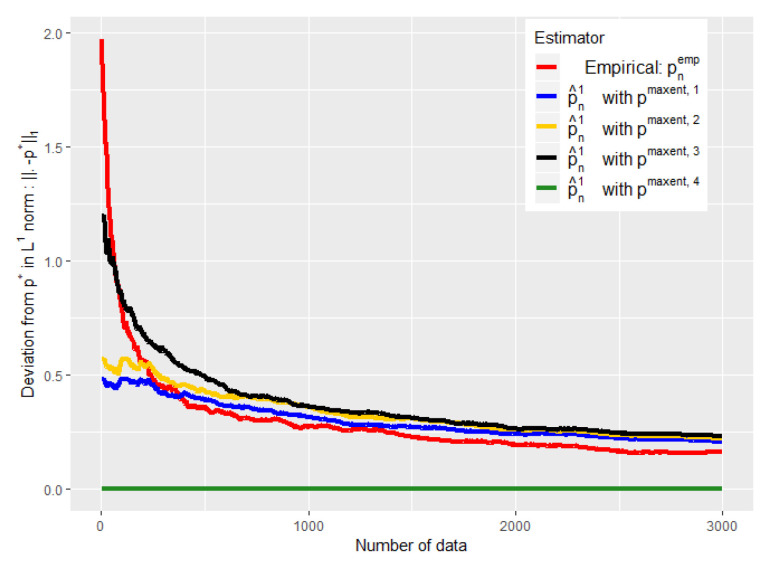
Evolution of the performance of ${\hat{p}}_{n}^{1}$ as a function of the available number of empirical data. $\epsilon_{n}$ is defined by Equation (Equation 20).

**Figure 5 entropy-21-01208-f005:**
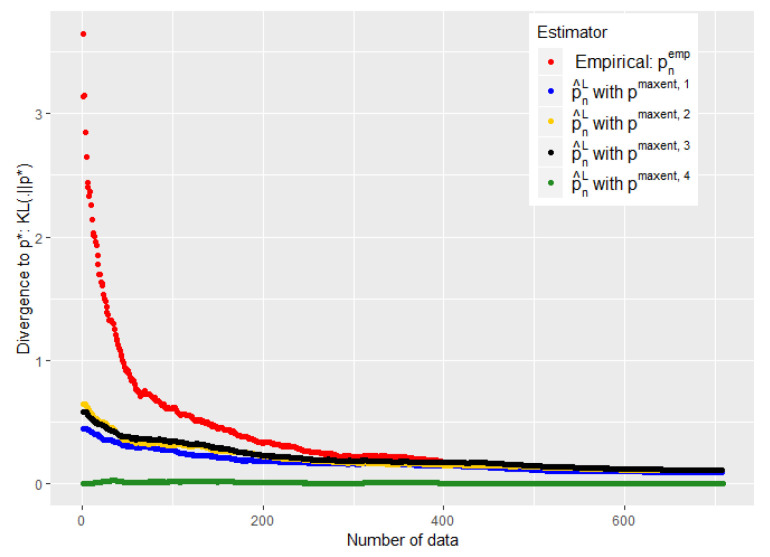
Evolution of the performance of ${\hat{p}}_{n}^{L}$ as a function of the available number of empirical data. $\epsilon_{n}$ is defined by Equation (Equation 19). Number of symptoms: 7.

**Figure 6 entropy-21-01208-f006:**
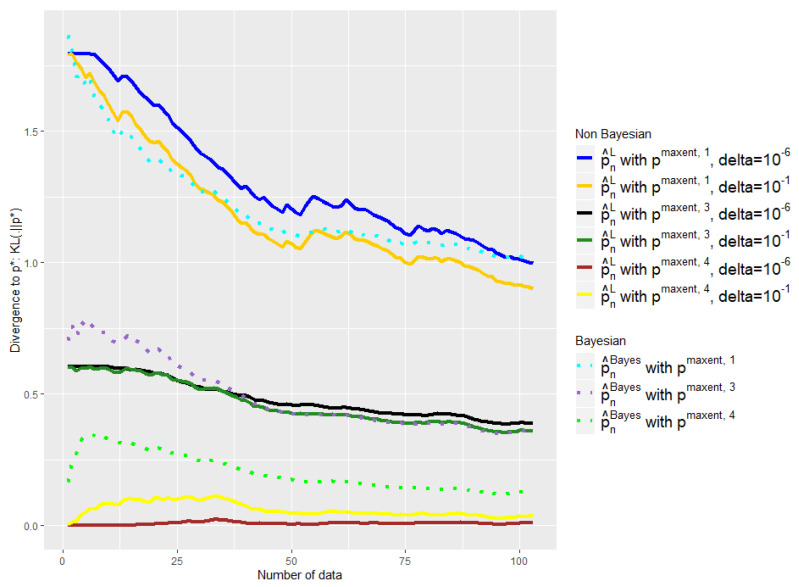
Comparison of the performances of ${\hat{p}}_{n}^{L}$ and ${\hat{p}}_{n}^{\mathrm{Bayes}}$ as a function of the available number of empirical data with different initial a priori and δ. $\epsilon_{n}$ is defined by Equation (Equation 18). Number of symptoms: 7.

**Figure 7 entropy-21-01208-f007:**
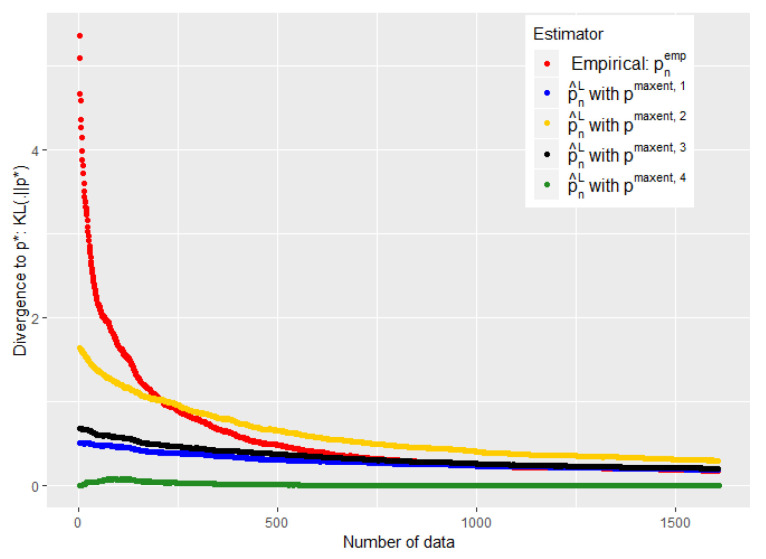
Evolution of the performance of ${\hat{p}}_{n}^{L}$ as a function of the available number of empirical data. $\epsilon_{n}$ is defined by Equation (Equation 19). Number of symptoms: 9.

**Figure 8 entropy-21-01208-f008:**
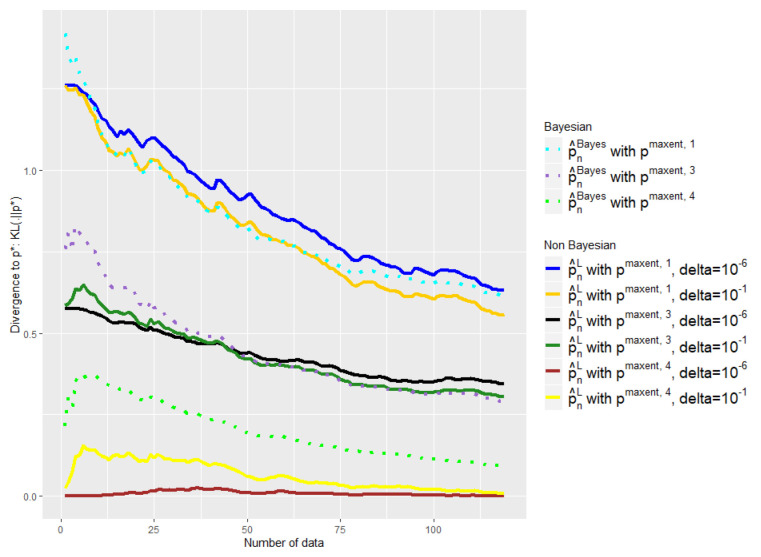
Comparison of the performances of ${\hat{p}}_{n}^{L}$ and ${\hat{p}}_{n}^{\mathrm{Bayes}}$ as a function of the available number of empirical data with different initial a priori and δ. $\epsilon_{n}$ is defined by Equation (Equation 18). Number of symptoms: 9.

**Figure 9 entropy-21-01208-f009:**
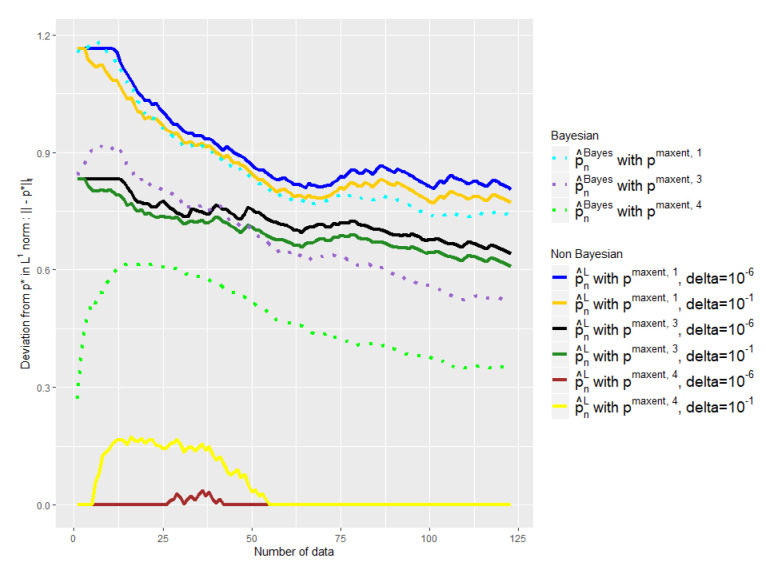
Comparison of the performances of ${\hat{p}}_{n}^{1}$ and ${\hat{p}}_{n}^{\mathrm{Bayes}}$ as a function of the available number of empirical data with different initial a priori and δ. $\epsilon_{n}$ is defined by Equation (Equation 20). Number of symptoms: 9.

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
