# Peer review of "Learning from Both Experts and Data"

_entropy, 2019, doi:10.3390/e21121208_

Round 1
Reviewer 1 Report
This paper aims to formulate a principled approach to estimating discrete probability distributions by combining expert-defined distributions (similar to a Bayesian prior distribution) with empirical distributions. The authors define a barycenter estimator which yields the distribution which is closest to the expert estimate, but still within a distance εn to the empirical distribution. This is defined in terms of an arbitrary distance measure, and the authors further formulate (a) normed spaces, and (b) the Kullback-Leibler divergence. In its theoretical definition and both choices of distance measure, computing the threshold distance εn appears to require knowledge of the true underlying distribution; however, in numerical practice it can be defined in terms of a parameter δ, and it is claimed that the results are not particularly sensitive to its choice. The authors claim that this barycenter distribution yields a better estimate than either the empirical or expert distributions alone, especially in intermediate regimes for which either the empirical or expert distributions are only partially accurate (e.g. when there is not very much data).
Combining expert prior knowledge with data is indeed a topic of widespread interest, and broadly applicable to a variety of domains for which only a small-to-intermediate amount of data is available. However, the merit of a novel approach depends upon how it performs relative to existing methods. To this end, I have the following concerns:
(1) It is unclear how accurate this method is compared to existing methods for combining expert knowledge with data. In some cases, this method appears to yield a worse estimate than just taking the empirical distribution directly (e.g. the yellow line in Figure 5). How would, for example, a Bayesian approach perform given the same data and the same prior? A direct comparison with existing methods should be performed to assess the utility of this approach.
(2) Figure 7 does appear to demonstrate that the choice of δ (and, equivalently, the value of εn) has only a small effect on the results, and this relative insensitivity to parameter choice is taken as one of the advantages of this approach. However, I am confused about how the results could be so insensitive given my understanding of the role of εn from earlier in the paper. Consider the two limits: for a sufficiently small choice of εn, we would choose a distribution arbitrarily close to the empirical distribution; conversely, for a sufficiently large choice of εn, we would always choose exactly the expert distribution. Is that correct? Can the authors comment on why we nonetheless see little sensitivity? δ is certainly calculated over a broad range -- but is this the appropriate range?
Minor Comments:
(3) Figures 1 and 2 have identical captions. Please differentiate between the figures and elaborate on what they're each showing within the captions themselves.
(4) Eq. 7: If this is motivated by Eq. 5, shouldn't p^expert and p^emp be flipped around here?
(5) The conclusion states "our barycenter estimator ... is always more efficient than the best of two models (clinical data or experts alone)." Is this consistent with Figures 5 and 7? For example in Figure 5, it appears that the Empirical distribution quickly becomes better than the yellow curve. What precisely is meant by "more efficient" here?
Reviewer 2 Report
Dear Authors,
see the attached file for a few remarks and suggestions.

Round 2
Reviewer 1 Report
I thank the authors for their quick work in generating additional results, and in answering my comments. I believe that the paper's claims are more soundly established given the new results, and that my questions and comments have been adequately addressed.